

# Technical Note: Bioaerosol identification by wide particle size range single particle mass spectrometry

**Xuan Li[1,2], Lei Li[1,2], Zeming Zhuo[1,2], Guohua Zhang[3], Xubing Du[1,2], Xue Li[1,2], Zhengxu Huang[1,2], Zhen Zhou[1,2] , and Zhi Cheng[4]**

[1]Institute of Mass Spectrometry and Atmospheric Environment, Jinan University, Guangzhou, 510632, China

[2]Guangdong Provincial Engineering Research Center for On-Line Source Apportionment System of Air Pollution, Guangzhou, 510632, China

[3]Guangzhou Institute of Geochemistry, Chinese Academy of Sciences, Guangzhou, 510640, China

[4]Institute of Systems Engineering, Academy of Military Sciences, Tianjin, 300161, China

**Correspondence:** Zhi Cheng(chengzh@npec.org.cn)

**Abstract** The sources of bioaerosols are complex and diverse, which have a direct impact on the environment, climate, and human health. The effective identification of bioaerosols in the atmosphere is greatly significant for accurately obtaining the atmospheric chemical characteristics of bioaerosols and making biological early warnings and predictions. To improve the identification ability of bioaerosols, this study detected a variety of bioaerosols and abiotic aerosols based on a single particle aerosol mass spectrometry (SPAMS). Furthermore, the bioaerosol particle identification and classification algorithm based on the ratio of phosphate to organic nitrogen was optimized to distinguish bioaerosols from abiotic aerosols. The results show that 15 kinds of pure fungal aerosols were detected by SPAMS based on a wide range sampling system and that fungal aerosols with a particle size up to 10 μm could be detected. Through the mass spectra peak ratio method of $PO_3^-/PO_2^-$ and $CNO^-/CN^-$, when discriminating abiotic aerosols, such as disruptive biomass combustion particles, automobile exhaust, and dust, from pure bacterial aerosols, the discrimination degree was up to 97.7%. The optimized ratio detection method of phosphate to organic nitrogen has strong specificity, which can serve as the discriminant basis for identifying bioaerosols in SPAMS source analysis or other analytical processes.

**Keywords** Bioaerosol; Single particle aerosol mass spectrometer (SPAMS); Online identification



## 1 Introduction


As a crucial component of atmospheric organic aerosols, bioaerosols participate in the
weather and climate process as cloud condensation nuclei and ice nuclei (Fröhlich-Nowoisky et al.,
2016). Moreover, some aerosols are human allergens, which pose a great threat to human health.
At present, the importance of bioaerosols (Burrows et al., 2009) has been fully recognized;
however, the sources of bioaerosols are sometimes difficult to identify, given their wide and
scattered sources (Li et al., 2021), in addition to the obvious influence of meteorological
conditions. This makes the identification of bioaerosols in the environment difficult (Rosch et al.,

42 2006).

At present, laser-excited fluorescence spectroscopy is widely employed to detect bioaerosols
(Li et al., 2018) due to its strong fluorescence signal, relative ease of operation, long-distance
identification of bioaerosols and abiotic aerosols, and determination of single-molecule particle
spectra. The fluorescent groups contained in bioaerosol particles are used for their detection in the
fluorescence spectrometry method. However, since some inorganic minerals also fluoresce under
ultraviolet light excitation, it is difficult to exclude the interference of abiotic fluorescent particles
in the identification process. For instance, polycyclic aromatic compounds or humic acids can
have similar fluorescence properties (Gabey et al., 2010) and cigarette smoke has similar
fluorescence properties to bacteria (Hill et al., 1999). In recent years, the single particle mass
spectrometry detection technology of bioaerosols has been developed rapidly, which can obtain
the particle size information and chemical composition of single particles in real-time online.
However, single particle aerosol mass spectrometry (SPAMS) also has its shortcomings in
identifying environmental bioaerosols (Kleefsman et al., 2007). As phosphorus and nitrogen are
components of nucleic acids and cell membranes, there is a large number of phosphate ions ($PO^-$,
$PO_2^-$, and $PO_3^-$) and organic nitrogen ions ($CN^-$ and $CNO^-$). Therefore, particles containing
phosphate and organic nitrogen in the ambient air (such as biomass combustion products (Wei et
al., 2019), fly ash, road dust (Yu et al., 2017), vehicle exhaust (Sodeman et al., 2005), and soil dust
(Silva et al., 2000) are often confused with bioaerosols in the detection process. To improve the
identification of bioaerosols, Zawadowicz et al. (2017) proposed a classification algorithm of
spectral peak ratio based on $PO_3^-/PO_2^-$ and $CN^-/CNO^-$. When using particle analysis by laser mass
spectrometry to discriminate dust and combustion by-products from pollen and bacterial aerosols,
the degree of confidence is up to 98%. However, the research on the detection and discrimination
of bioaerosols from fungi and other bioaerosols remains insufficient.
In addition, the particle size distribution of bioaerosols is generally 0.3–100 μm, while that of
viruses is less than 0.3 μm (Smets et al., 2016). The typical particle size of bacteria is 0.25–8 μm,
while that of fungi is 1–30 μm, among which the particle size that can cause harm to the human



body is generally between 0.4–10 μm. For most of the existing SPAMS, the particle size analysis
ability is about 0.1–3 μm, thus the ability of SPAMS to detect fungi, spores, and other large
particles is limited. Williams et al. (2013) designed a 7-stage aerodynamic lens (A-lens) to
improve the ability of aerodyne aerosol mass spectrometry to detect biological particles. By
optimizing the buffer cavity and increasing the sampling pressure of the lens, the transport
efficiency of aerosol particles in the size range of 200–5000 nm can reach 100% but that of 10 μm
particles is only 22%. At the same time, Cahill et al. (2014) constructed a 7-stage A-lens for the
transmission of a single particle in the range of 4–10 μm and found that the transmission
efficiency of 10 μm particles was less than 20%, while that of 3 μm particles was close to 0.
ATOFMS is attempted to extend to the study of single-cell metabolomics.
To improve the ability of SPAMS to detect bioaerosols, a new sampling system (wide particle
size range is 0.1–10 μm) was verified in the preliminary design. The particle transport efficiency
can reach 100% in the particle size range of 0.15–10 μm. In this study, the advantages of high
performance-SPAMS (HP-SPAMS) in the detection of bioaerosols were explored and 10 μm
fungal particles were successfully detected. Furthermore, the discrimination method based on the
ion ratio of bioaerosol characteristic peak was further optimized and verified, which could
successfully discriminate bioaerosols from common disruptors, such as road dust, vehicle exhaust,
and biomass combustion products, with a discrimination degree of 97.7%. In addition, this
discrimination method had good statistical significance for bacterial aerosols. The analysis of
single particle mass spectra is a hard ionization process and laser energy has little effect on the
discrimination of this classification method. This method can be used as a discriminant basis for
identifying bioaerosols in SPAMS source analysis or other analyses.

## 2 Experiment


### 2.1 SPAMS


The constitution and working principle of SPAMS have been described in detail by Li et al.
(2011). In short, SPAMS uses an aerodynamic lens to introduce aerosols from the atmosphere into
a vacuum system and focus them into a collimating particle beam. Two successive laser beams are
then used to measure the flight speed of particles and calculate their aerodynamic diameter in turn.
The high-energy pulsed laser ionizes particles into positive and negative ions at the center of the
ion source, which is detected by time-of-flight mass spectrometry.
In this study, the performance of HP-SPAMS was improved on the original basis and the
resolution of the time-of-flight mass spectrometer was improved using delayed extraction
technology (Li et al., 2018). The multi-channel superimposed signal acquisition system improves
the sensitivity and dynamic range of instrument detection (Shen et al., 2018). In particular, for the



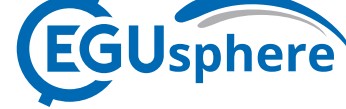

improvement of the sampling system (Zhuo et al., 2021), the whole sampling system consists of
five modules: pre-focusing sampling interface, virtual impact concentration device, buffer cavity,
7-stage aerodynamic lens, and acceleration nozzle. The pore size was expanded from 0.11 μm to
0.22 μm. The numerical results show that the theoretical transport efficiency of particles could
reach 100% in the particle size range of 0.15–10 μm, which improved the sampling capacity of
SPAMS for coarse particles and provided conditions for the detection of bioaerosols.
**2.2 Sample Determination**
The 10 strains of bacteria and five strains of fungi determined in this study were standard
strains provided by Hexin Kangyuan Medical Technology Co., Ltd. (Guangzhou). The specific
names are shown in Table 1. Biological aerosol disruptors often found in the real environment,
such as road dust, vehicle exhaust, and biomass combustion products (wheat stalk, corn stalk, and
oblate leaf stalk), were selected as the research objects.
The preparation steps of the strain sample solution are as follows: 1) Sampling: first, the
strains refrigerated at -80°C were taken out, thawed at room temperature for 1–2 hours, and
vortexed using an oscillator to shake the centrifugation tube of the strain sample evenly. 2)
Inoculation: on a clean laboratory table, the strain solution adhered to the disposable sterilized
inoculating loop, and streak inoculation on the blood agar plate medium was performed. 3)
Culture: the streaked culture medium was placed horizontally in a 37°C constant temperature box
for about 24 hours. 4) Sampling: the growth of the samples of 15 strains after the culture is shown
in Table 1. The colonies on the surface of the blood agar were slightly scraped with a disposable
sterilized inoculating loop, dissolved with 1 mL deionized water in the centrifuge tube, and shaken
well. 5) Dilution with water-soluble salt: the bacterial sample aqueous solution was centrifuged for
3 min at the rotation speed of 3,000–5,000 rpm. After centrifugation, the sample was precipitated
at the bottom of the centrifuge tube and the aqueous solution was absorbed. Then, 1 mL of
deionized water was added to dissolve the precipitate, followed by thorough shaking. Step 5 was
repeated thrice.
The main components of blood agar plate medium used in this experiment were peptone, beef
powder, sodium chloride, defiber sheep's blood, agar and deionized water. All media were
autoclaved prior to use. Scrape only the upper layer of the culture medium surface to avoid small
contaminants of the culture medium itself. Repeated rinsing with deionized water removes excess
salt. It should be stressed that no additional fixatives or epoxies were added to the cells before
analysis, reducing complications in the interpretation of the mass spectra.
The prepared pure bacterial sample solution was mixed with 20 mL of deionized water and
atomized using a single nozzle aerosol generator (TSI Inc., Model 9302) to obtain the aerosol
particles of the samples. A sheath gas of 80 kPa of clean air was used. The atomized sample





aerosol was connected to a silica gel drying tube, whose outlet was connected to the SPAMS inlet
and an exhaust port with a high-efficiency particulate air filter. When sampling, 1,000 effectively
ionized particle size, and spectrum data were stored in each sample. The experimental flow of
HP-SPAMS is shown in Fig. 1.


Table 1 Sample numbers and names of the 15 strains

| number of samples | name | state of bacteria or fungal |
|---|---|---|
| #01 | Klebsiella pneumoniae | |
| #02 | Salmonella pneumoniae | |
| #03 | Shiga virulent Escherichia coli | |
| #04 | Bordetella bronchitis | |
| #05 | Escherichia coli | |
| #06 | Staphylococcus aureus | |
| #07 | Listeria monocytogenes | |
| #08 | Enterococcus faecium | |
| #09 | Enterobacter cloacae | |
| #10 | Staphylococcus epidermidis | |
| #11 | Candida albicans | |
| #12 | Candida tropicalis | |
| #13 | Candida glabrata | |
| #14 | Aspergillus brasiliensis | |
| #15 | Saccharomyces cerevisiae | |




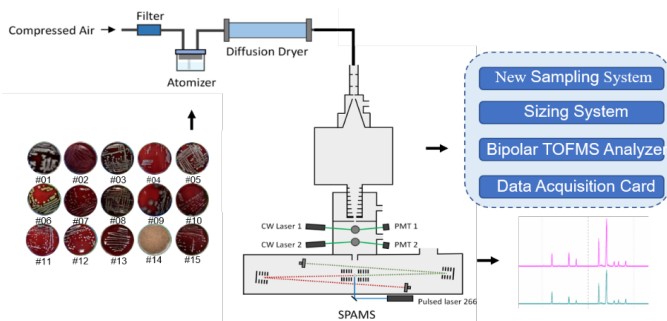


Fig. 1 Experimental flow of HP-SPAMS

## 3 Results and Conclusions

### 3.1 Distribution of bioaerosol particle size

The vacuum aerodynamic particle size distribution of the bacterial and fungal particles is
shown in Fig. 2. Preliminary experimental results showed that the overall particle size of bacteria
was relatively smaller, except for #08 *Enterococcus faecium*. The overall particle size distribution
of bacteria was mainly within the range of 0.3–1 µm, thus showing an approximately normal
distribution. Jung and Lee (2013) used scanning electron microscopy to observe *Escherichia coli*
and *Bacillus subtilis* cells at room temperature, both bacteria with a diameter in the range of 0.5–
0.7 µm and a length in the range of 1.1–1.6 µm. Fungi samples (#11, #12, #13, #14, and #15)
obtained from pure strain cultures had a much larger proportion of particle size distribution above
1 µm than bacteria, and the particle size distribution of three fungi, *Candida albicans*, *Candida*
*glabrata*, and *Saccharomyces cerevisiae* were around 0.5–2 µm. Sample #13 *C. glabrata* was
concentrated in the range of 1–2.5 µm. Compared with other samples, the samples #11 *C. albicans*
and #12 *Candida tropicalis* were similar to the samples #14 *Aspergillus brasiliensis* and #15 *S.*
*cerevisiae* in terms of particle size distributions, while the particle size distribution of samples #11
and #15 was mainly in the range of 0.25–1.5 µm and the peak was around 0.4 µm. It is worth
noting that the particle size of *C. tropicalis* and *A. brasiliensis* were evenly distributed between 0.1
µm and 10 µm. Li et al. (2020) used transmission electron microscopy and scanning electron
microscopy to investigate primary biological aerosol particles (PBAPs) collected from boreal
coniferous forests in the Xiao Hinggan Mountains of China in summer and speculated that the size
of rod PBAPs was distributed at 1.4 µm and 3.5 µm and that the two typical peaks were bacterial
and fungal particles, respectively.
Due to the low transmission efficiency of the aerodynamic lens for large particles and the
tendency of large particles to produce inertial impinging wall loss in the process of air transport,
especially large particles above 1 µm, the detected particle size distribution was smaller than the
real one. Therefore, for the first time, HP-SPAMS was used to measure 10 µm coarse particulate



matter under the improvement of the sampling system.

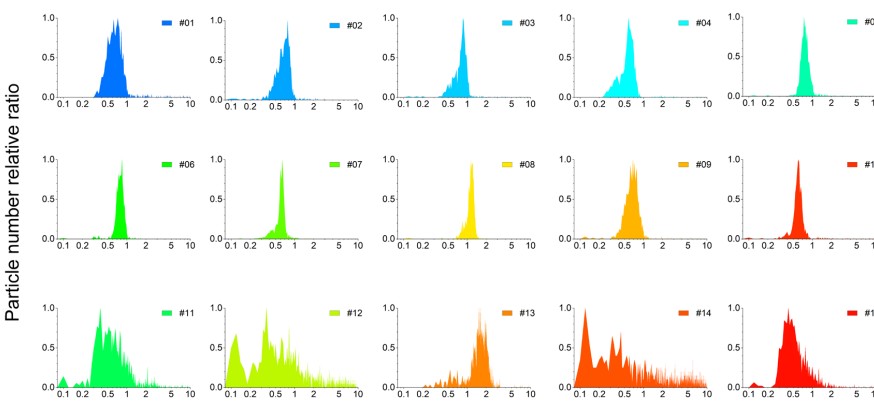


Fig. 2 Vacuum aerodynamic size distribution of 15 biological samples detected by SPAMS

**3.2  Characteristic spectrum of bioaerosols**

Zeng et al. (2019) used SPAMS to detect 13 strains of bacteria in 2019 to obtain similar

bioaerosol characteristic ions; however, there were fewer characteristic peaks in the negative mass
spectrum, and no negative ions with a mass charge ratio greater than 200 were detected, and the
overall ionic peak signal was 50–10,000. HP-SPAMS detected the mass spectra of 15 strains of
pure bacteria as shown in Fig. 3. Similar to the SPAMS detection results of the same type in the
world, they all could effectively measure the phosphate and organic nitrogen ionic peaks of active
bacterial aerosols, as well as some amino acid decarboxylic ionic peaks in the positive mass
spectrum. Czerwieniec et al. (2005) found similar peaks when detecting vegetative cells of
*Bacillus atrophaeus* and speculated that +30, +70, +72, +74, +86, +110, and +120 were
decarboxylic ionic peaks of amino acids. The positive ion peaks were mainly $^{30}$[Glycine-COOH]$^+$,
$^{70}$ [Proline-COOH]$^+$, $^{72}$ [Valine-COOH]$^+$, $^{74}$ [Threonine-COOH]$^+$, $^{86}$ [Leucine-COOH]$^+$,
$^{110}$ [Histidine-COOH]$^+$, and $^{120}$ [Phenylalanine-COOH]$^+$. Srivastava et al. (2005) speculated that
+59, +81, +84, and +88 ionic peaks were organic fragments containing nitrogen, among which
$^{84}$ [C$_5$NH$_{10}$]$^+$ with a strong signal was also found in the detection of this study. The negative ionic
peaks were mainly organic nitrogen $^{26}$CN$^-$, $^{42}$CNO$^-$, phosphate $^{63}$PO$_2^-$, $^{79}$PO$_3^-$, $^{97}$H$_2$PO$_4^-$,
$^{159}$H(PO$_3$)$_2^-$, $^{199}$ NaH$_2$P$_2$O$_7^-$, and other common biological ionic peaks.

More abundant ion characteristics were obtained by HP-SPAMS. On the original basis, the

decarboxylic ionic peaks of serine and alanine $^{44}$ [Alanine-COOH]$^+$ and $^{60}$ [Serine-COOH]$^+$ were
supplemented and the signal intensity in the HP-SPAMS detection results was relatively strong.
The negative ion peaks $^{261}$NaH(PO$_3$)$_3^-$ and $^{277}$NaH(PO$_3$)$_2$(PO$_4$)$^-$ with a mass charge ratio greater
than 250 were speculated and added. Exponential pulse delayed extraction technology (Chen et al.,





2020) not only solves the hit rate and resolution problems of SPMS but also improves the ion
signal intensity by multiple times, thereby providing conditions for obtaining the complete mass
spectrum characteristics of bioaerosols. More characteristic peaks can make it easier to distinguish
whether or not a single particle is a bioaerosol.

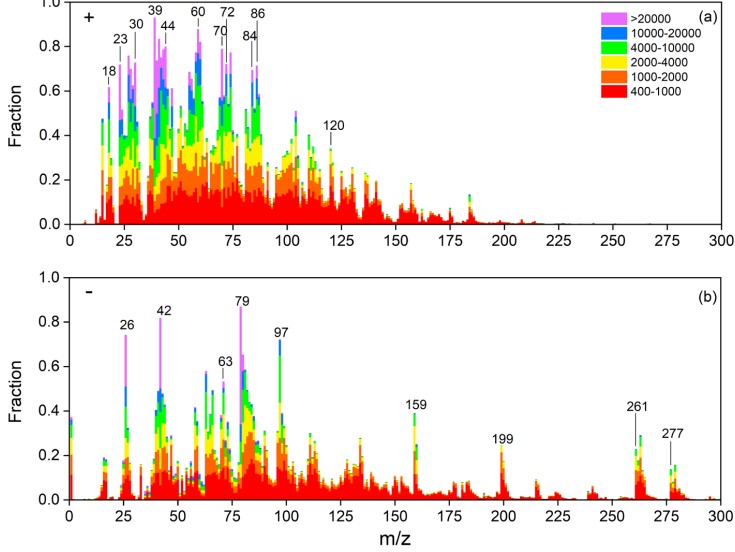


Fig. 3 Stacking diagram of the area of all bioaerosol ion peaks; (a) Positive mass spectrum; (b) Negative mass
spectrum
**3.3 Bioaerosol identification based on characteristic peak ratio**
HP-SPAMS can accurately screen out bioaerosols according to their characteristic ions.
However, due to the different intensity of ion signals in the single-particle spectrum, when the
signal of the characteristic peak is weak, the spectrum will be ignored and the bioaerosols cannot
be completely extracted. Zawadowicz proposed that $^{26}CN^-$, $^{42}CNO^-$, $^{63}PO_2^-$, and $^{79}PO_3^-$ could be
used as characteristic peaks of bioaerosol discrimination to distinguish bioaerosols from abiotic
aerosols in a larger proportion. In the actual environment, many inorganic particles contain
bioaerosol characteristic ion phosphate and organic nitrogen peaks. Organic nitrogen and
phosphate ionic peaks with strong signals also appear in biomass combustion products, vehicle
exhaust, and road dust measured by HP-SPAMS, as shown in Fig. 4. Therefore, when using
phosphate and organic nitrogen ionic peaks alone to identify bioaerosols in the environment, at
least 89% of vehicle exhaust, 49.5% of dust, and 58.3% of biomass combustion products have
interference, which cannot be directly used as a sufficient condition to distinguish bioaerosols.




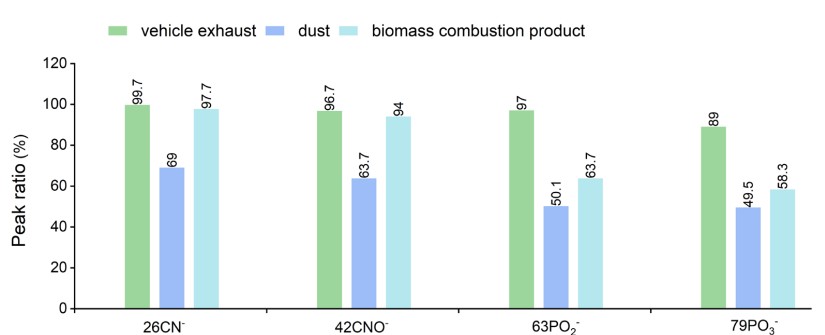


Fig. 4 Comparison of the frequency of four ion peaks in abiotic aerosols

Data classification based on HP-SPAMS, which is the peak height ratio of the mass spectra
peak of $PO_3^-/PO_2^-$ and $CNO^-/CN^-$, was used to distinguish bioaerosols from their disruptors. The
scatter distribution of $PO_3^-/PO_2^-$ and $CNO^-/CN^-$ was obtained by capturing the corresponding ionic
peak height. As shown in Fig. 5, the distribution of bioaerosols was significantly different from
that of biomass combustion products, vehicle exhaust, and dust. In particular, the scatter
distribution positions of $PO_3^-/PO_2^-$ and $CNO^-/CN^-$ were significantly different from that of vehicle
exhaust. Moreover, it was concentrated in a certain range of values. Bioaerosols were classified as
one class, while aerosols produced by vehicle exhaust, dust, and biomass combustion products
were classified as another class. The proportion interval of the bioaerosols $PO_3^-/PO_2^-$ and
$CNO^-/CN^-$ measured by SPAMS were concentrated at (3,200) and (0.7, 7), respectively, while
those of abiotic aerosols were at (0.2, 3) and (0.02, 2), respectively. Furthermore, using the support
vector machine (SVM), a supervised machine learning algorithm, the discrimination degree
between bioaerosols and abiotic aerosols was up to 97.7%. This indicates that HP-SPAMS had a
strong detection specificity on the phosphate and organic nitrogen ratio between bioaerosols and
abiotic aerosols. It is possible to use it as the discriminant basis for identifying bioaerosols in the
SPAMS source analysis or other analyses. Compared with the traditional single discrimination
method via life characteristic elements, such as nitrogen and phosphorus, this method has a higher
discrimination degree.





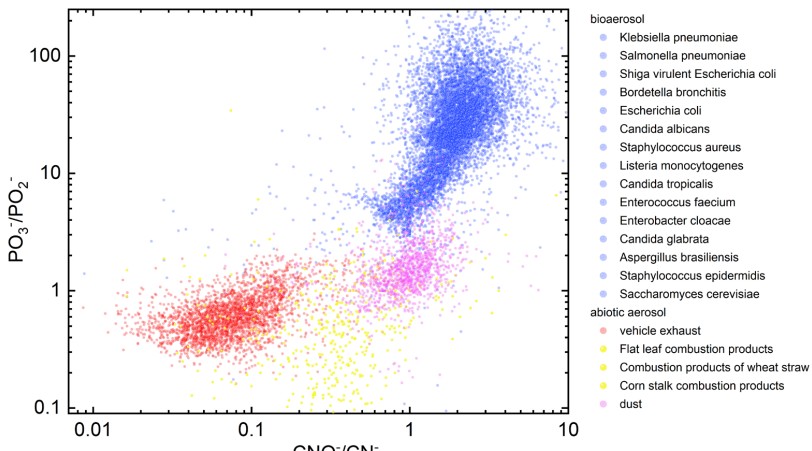


Fig. 5 Scatterplot of the bioaerosol and abiotic aerosol $CNO^-/CN^-$ and $PO_3^-/PO_2^-$


The premise of this classification method is that there are four characteristic peaks. However,
the actual research showed that when the threshold of the effective peak was set high, part of the
weak signal peaks would be filtered out; however, when the threshold was set low, there was noise
interference in the collected signals. Through a series of equivalent gradient threshold settings, the
effective peak threshold of 10 mV was determined. The average frequency of the characteristic
peaks in the bacterial aerosols was generally higher than that of the fungal aerosols, as shown in
Table 2. At least, 82.9% of bacterial aerosols and 52.8% of fungal aerosols could be effectively
discriminated against. The discrimination method based on the characteristic peak ratio had a high
identification rate for bacterial aerosols. Fungi and bacteria have great differences in terms of
morphology and structure. Bacteria are mainly coccus, bacillus, and spiral, while fungi are mainly
subcellular and multicellular. In addition, fungi have nuclei. In the detection process, bacterial
aerosols are more easily ionized to produce effective mass spectra peaks.
Table 2 The average frequency of the characteristic ionic peak of bioaerosol

| Species | $CN^-$ | $CNO^-$ | $PO_2^-$ | $PO_3^-$ |
|---------|--------|---------|----------|----------|
| Bacteria | 92.9% | 96.5% | 82.9% | 97.6% |
| Fungus | 63.8% | 70.4% | 52.8% | 75.3% |

**3.4  Influence analysis of laser energy**
SPAMS single-step laser desorption is a difficult ionization process in which organic
compounds produce ion fragments of different degrees. Noble et al. (1996) proposed that in single
particle mass spectrometry analysis, it is difficult to determine the morphology of organic
compounds due to the extensive fragmentation caused by the ionization process. Cornwell et al.



(2022) proposed that the ion signals of dust and biological particles were very sensitive to
ionization conditions and that total positive ion intensity was used to characterize the mass
spectral relationship between different dust and biological particles. Through this method,
environmental particles with both dust and characteristic biological spectra fingerprints were
successfully excluded from the classification of biological particles. Liu et al. (2021) used *Bacillus*
*thuringiensis* to explore the influence of different laser pulse energies on SPAMS and found that
particles did not ionize when the laser energy was lower than 0.2 mJ and that the ionic peak
increased significantly when the laser energy was higher than 1.5 mJ; they also found that the
ionic peak integrity was the best when the laser energy was about 0.5 mJ. Too high or too low
energy was not conducive to the discovery of the characteristic mass spectrum. To verify the
influence of ionized laser energy on this analysis method, different laser energies of 0.5, 0.75, 1.0,
1.25, and 1.5 mJ were selected and *Staphylococcus aureus* was taken as an example to explore the
influence of ionized laser energy on the ionic peak ratio. As shown in Fig. 6, as the energy
increased, the ionization degree increased with more fragmented ions, the ratio of $PO_3^-/PO_2^-$ and
$CNO^-/CN^-$ decreased, and more $PO_3^-$ and $CNO^-$ were ionized as $PO_2^-$ and $CN^-$, respectively.

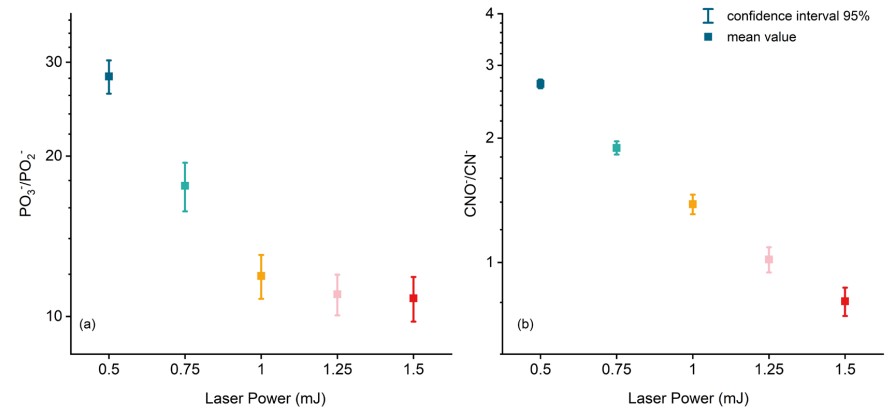

Fig. 6 Distribution of $PO_3^-/PO_2^-$ (a) and $CNO^-/CN^-$ (b)

Furthermore, the discrimination degree of ionized laser energy on bioaerosols under different
ratios was compared. The biological aerosol (*S. aureus*) and abiotic aerosol (dust) were selected
under the laser energy of 0.5 mJ. Using SPAMS, it was found that the ratio interval of $PO_3^-/PO_2^-$
and $CNO^-/CN^-$ was concentrated in (5, 70) and (0.8, 3), while that of dust aerosol was (0.7, 3) and
(0.5, 2), respectively. When the laser energy was 1.5 mJ, the ratio interval of bioaerosols was (2,
30) and (0.7, 3), while that of dust aerosols was (0.6, 3) and (0.6, 2). As shown in Fig. 7, The
interval of biological aerosols was gradually changing and the trends of horizontal and vertical
coordinates were both decreasing, while the scatter interval of abiotic aerosols was almost
unchanged. According to the SVM algorithm, the discrimination degree of bacterial aerosols and





dust under 0.5, 0.75, 1.0, 1.25, and 1.5 mJ energies were 96.6%, 97.4%, 97.1%, 96.5%, and 97.8%,
respectively, indicating that the ionized laser had little effects on discriminating biological aerosols
and dust disruptors.

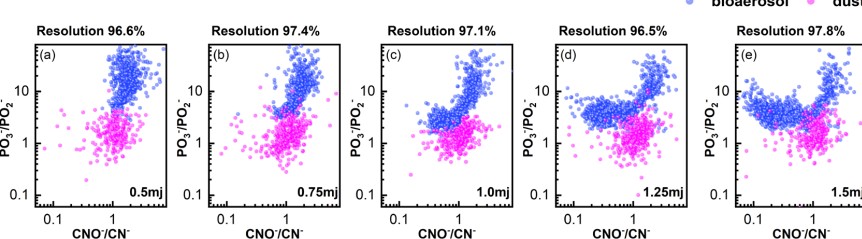

Fig. 7 Scatter distribution of the bioaerosol and dust aerosol $CNO^-/CN^-$ and $PO_3^-/PO_2^-$

Under the condition of a constant effective peak threshold, the frequency of phosphate and

organic nitrogen ionic peaks of bacterial aerosol and dust changed with the change of the ionized
laser energy, as shown in Fig. 8. When the laser energy was 0.5 mJ, the peak output rate of both
bioaerosol and dust was the lowest and the influence on abiotic aerosols was larger, with a peak
output rate of 34%. When using this classification method for discrimination, it is only necessary
to discriminate against 28.7% of dust particles. When the laser energy was 1.5 mJ, the peak output
of four *S. aureus* ionic peaks was the highest, while that of the dust was the lowest. At this time,
the highest proportion of bacterial aerosols (94.6%) and the lowest proportion of dust particles
(31.1%) could be statistically discriminated against. Under the same laser energy, the overall peak
output of abiotic aerosols was about 40% lower than that of biological aerosols. Different types of
particles had different laser energy requirements. In addition, the variation trend of $CNO^-$ and $PO_3^-$
was the same as that of $CN^-$ and $PO_2^-$, respectively; however, the phosphate ionic peaks ($PO_3^-$ and
$PO_2^-$) were more affected by the laser energy. In conclusion, when the ionized laser energy was 1.5
mJ, the classification method of the ionic peak ratio was more effective in discriminating
bioaerosols.



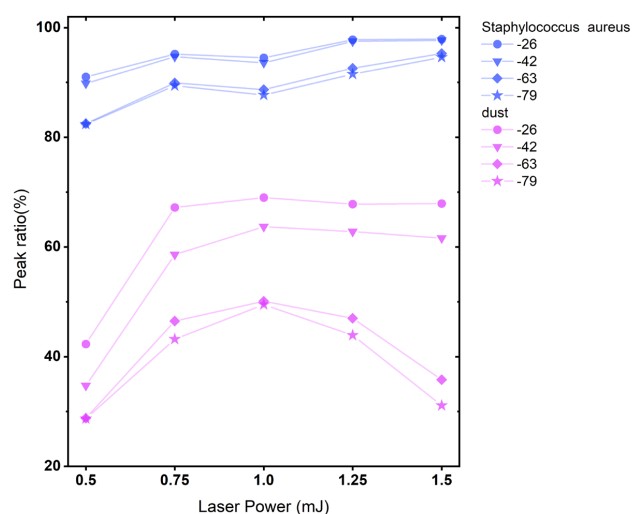


Fig. 8 Peak rate diagram of the characteristic peak output under different laser energies


## 4 Conclusions


The performance of SPAMS and the improvement of the sampling system have improved the
ability to identify bioaerosols. HP-SPAMS was used for the first time to detect fungal particles
with a particle size of 10 μm, which provided a good technical basis for the detection of biological
aerosols with large particle sizes in the environment. With the improvement of the instrument
performance, the single-particle spectrum of bioaerosol showed decarboxylic ionic peaks of serine
and alanine $^{44}$[Alanine-COOH]$^{+}$ and $^{60}$[Serine-COOH]$^{+}$ and phosphate ionic peaks $^{261}$NaH(PO$_3$)$_3^{-}$
and $^{277}$NaH(PO$_3$)$_2$(PO$_4$)$^{-}$. A more unique fingerprint spectrum than the original study was obtained.
The bioaerosol identification method based on the characteristic peak ratios PO$_3^{-}$/PO$_2^{-}$ and
CNO$^{-}$/CN$^{-}$ can effectively discriminate bioaerosol from three kinds of commonly seen abiotic
disruptors, with the discrimination degree up to 97.7%. In addition, due to the influence of laser
ionization efficiency, the effective mass spectra peak ratio of bacterial aerosol generation is higher,
thus it is more suitable for this method. The ionized laser energy has a certain influence on the
integrity of the ionic peak but hardly affects the identification accuracy of bioaerosols. This study
showed that the SPAMS detection technology of bioaerosols has the potential to be a new method
for real-time online identification of bioaerosols.

Data availability. These data can be publicly accessible in free.

Author contributions. LL and ZC designed the study; XL and ZMZ performed the experiments;
GHZ, XBD, XL, ZXH and ZZ participated in data analysis and result discussion; XL and LL



wrote the paper with the input from all authors.

Competing interests. The authors declare that they have no conflict of interest.

Acknowledgements. We would like to thank engineer Huang Fugui of the Guangzhou Hexin Mass
Spectrometer Co., Ltd., for his technical support .

Financial support. This research has been supported by the National Natural Science Foundation
of China (grant no. 41905106).

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
