# Peer review of "Technical Note: Bioaerosol identification by wide particle"

_EGUsphere, 2022_

## Author Comment (AC1)

**Reply on RC1**

The authors have explained the general background and previous research on the topic well. However, the last paragraph of introduction are results from the study and should be moved to results and discussion. The aims of the study are not clearly stated and should be included in the introduction section. In the results section, it was sometimes hard to differentiate what was literature review and what were the new results reported by the authors.

- Add details for choosing these 15 strains

**Answer:** Thank you for your good suggestion. The 15 strains, which include bacteria, mold and yeast, are representative. The bacteria included both Gram-positive and Gram-negative bacteria, and included common shapes such as balls and rods. We have added the types of bioaerosols in Table 1.

- State the reasons for choosing blood agar and the incubation periods for both bacteria and fungi. Fungi often takes longer to grow in agar compared to bacteria.

**Answer:** This is really a very professional question, thank you very much. Blood plate medium is rich in nutrients, which can meet the growth requirements of strains and facilitate the separation of samples. The streaking bacterial medium was placed horizontally in a 37°C incubator for 18-24h, and the fungal medium was placed horizontally in a 25°C incubator for 36-48h. We have supplemented the section in sample preparation.

- In section 3.1, how did the authors determine if the biological particles were single cells or clumps?

**Answer:** This is really a very professional question, thank you very much. Single particle aerosol mass spectrometry is a technique in which the particle size is obtained by converting the velocity of particles measured by double-beam diameters to the calibration curve measured by standard monodisperse PSLs pellets. By strictly controlling the injection concentration of bioaerosol to 80±10 per second, the phenomenon of "particle catch-up" caused by excessive aerosol concentration was excluded. And the best way to ensure the existence of single-cell particles is to generate bioaerosol through nebulizer. We compared the size distributions of the bioaerosols detected by HP-SPAMS with those obtained by electron microscopy. HP-SPAMS, like all single-particle mass spectrometers have size-dependent counting biases in the range of 10% compared with electron microscopic measurements (for example E. coli). But there is no guarantee that all the particles tested will be single cells.

- Is it probable to add bioaerosols to fig 4 ?

**Answer:** Thank you for your good suggestion. We added bioaerosols in Fig.4 to better compare the ion signals of the four types of particles in organic nitrogen and phosphate.

[Figure]

Fig. 4 Comparison of the frequency of four ion peaks in abiotic aerosols

- Table 2 reports only average. It would be more informational with average±SD

**Answer:** Thank you for your suggestion. To provide more specific information, we supplement average ±SD in Table 2.

Table 2 The average frequency of the characteristic ionic peak of bioaerosol

| Species | $CN^-$ | $CNO^-$ | $PO_2^-$ | $PO_3^-$ |
|---------|--------|---------|----------|----------|
| Bacteria | 92.9±1.8% | 96.5±1.1% | 82.9±5.0% | 97.6±2.5% |
| Fungi | 63.8±21.1% | 70.4±21.3% | 52.8±18.5% | 75.3±26.6% |

- In section 3.4, how representative is S. aureus for all bioaerosols? The authors could add a representative fungus or a mixture of bacteria and fungi.

**Answer:** Thank you for your good suggestion. Staphylococcus aureus is widely found in the nature environment. The characteristics of the 15 strains were analyzed by mass spectrometry, and the similarity between the strains was very high. The characteristics of S. aureus spectra are highly consistent with the measured all bioaerosols characteristics and have certain representativeness.

[Figure]

- Page 4 – The tense is inconsistent for sample determination section.

**Answer:** Thanks. It has been corrected.

- Line 195 – Serine and alanine needs to be formatted.

**Answer:** Thanks. We have made the formatting changes.

- Table 2 – Species "Fungus" is singular. I believe the authors are referring to a group of fungi.

**Answer:** Thank you, and "Fungus" has been changed into "Fungi".

---

## Author Comment (AC2)

**Reply on RC2**

The Technical Note, "Bioaerosol identification by wide particle size range single particle mass spectrometry" by Li, et al. describes a bioaerosol identification algorithm for a single-particle mass spectrometer SPAMS. The details of the classification are the same as those in Zawadowicz, et al. (2017) (ratios of characteristic phosphorus and organic nitrogen markers discriminated in biological and abiotic species by use of the SVM algorithm), but this is a valid contribution to the literature because it reproduces the previous study on a different single-particle instrument with a different training dataset. Particularly, it is of note that SPAMS and PALMS use ionization lasers of different wavelengths, which would be the most significant reason for this method to work with one instrument and not the other. In this way, this study is also somewhat contrary to the recent Cornwell et al. (2022) paper, which argues that the phosphate-ratio SVM technique is too sensitive to laser ionization energy to work on the ATOFMS system, which uses the same ionization wavelength as SPAMS.

Overall, support the publication of this paper after the authors address the following comments. I will rely on the editor's best judement whether this paper would fit better at ACP or AMT. My first instinct was to recommend resubmission to AMT, but I am not very familiar with the scope of the new Technical Note manuscript type, and reading ACP's guidelines it seems like it fits with "development of numerical algorithms for the interpretation of atmospheric data (such as statistical methods and machine learning)."

Overall comment: the paper would benefit from another read for syntax, grammar, etc. Some sentences are difficult to follow (some, but not all instances are noted below).

- p.2, line 64-65: Not sure I follow this sentence, "However, the research on the detection and discrimination of bioaerosols from fungi and other bioaerosols remains insufficient". This reads like you are suggesting the need for increased speciation of bioaerosol detected by SPMS (i.e. discriminating fungal bioaerosols from other bioaerosol types), but this is not further discussed in the paper, which focuses on discriminating between bioaerosol and abiotic aerosol.

**Answer:** This is really a very professional question, thank you very much. We detected bacterial bioaerosols and fungal bioaerosols by SPAMS, but they could not be distinguished according to their mass spectral characteristics. The paper focuses on discriminating between bioaerosols (bacteria and fungi) and abiotic aerosols. In the process of literature review, we found that there are few studies on the distinction between fungi and abiotic aerosols. Here, the sentence was

changed as "**However, there is insufficient research on the detection and differentiation of other bioaerosols such as fungi from abiotic aerosols**".

- Experimental section: can you provide some rationale for choosing these specific bacterial and fungal strains?

**Answer:** Thank you for your good suggestion. The 15 strains, which include bacteria, mold and yeast, are representative. The bacteria included both Gram-positive and Gram-negative bacteria, and included common shapes such as balls and rods.

- Bacterial and fungal strains that form the training dataset are discussed in the Experimental section, but road dust, exhaust and combustion products are not. Please provide some discussion of what types of abiotic phosphorus-containing species were used in the study.

**Answer:** Thank you for your good suggestion. The specific types of abiotic nitrogen and phosphorus species discussed in this manuscript are described in the following table.

Table 1 Summary of abiotic species samples

| Sample | Description | Study |
|---|---|---|
| road dust | Guangzhou Accelerator Industrial Park road dust | Reported for the first time |
| Vehicle exhaust | Fresh exhaust collected from a light-duty diesel vehicle with the engine started and at steady state | Su et al.,2021, Journal of Hazardous Materials. |
| wheat stalk combustion products | Stems and leaves of mature wheat in East China | Reported for the first time |
| corn stalk combustion products | Stems and leaves of mature corn in East China | Reported for the first time |
| oblate leaf combustion products | Dried oblate leaves of eastern China | Reported for the first time |

- p. 6, line 173: "Therefore, for the first time, HP-SPAMS was used to measure 10 μm coarse particulate matter under the improvement of the sampling system." Can you provide some figures of merit for your new coarse-mode sampling system? What is the transmission efficiency at 10 μm?

**Answer:** Thank you for your interest in the new coarse-mode sampling system. Fig. 1 shows a schematic diagram of the wide-range inlet system, which consists of four modules: a pre-focusing sampling connection, a virtual impact sampling cone, a relaxation chamber, and a seven-stage aerodynamic lens. Figs. 1b, 1c, and 1d show the detailed dimensional parameters of each module. Fig.2 shows the theoretical and experimental comparison of particle transmission efficiency provided by the wide-range aerosol inlet system. In the experiment, the pulse numbers of the two PMT beams were counted respectively. $PMT_1$ and $PMT_2$ were 12 cm and 18 cm away from the outlet of the lens system, respectively. The transmission efficiency of the lens system could be roughly estimated by the pulse number of $PMT_1$. In the entire particle size range of 120 nm-10 μm,

the transmission efficiency calculated through PMT$_1$ counting was consistent with the theoretical simulation. In the particle size range of 200 nm-10 μm, the transmission efficiency of PM was close to 100%. Overall, when the wide-range aerosol inlet system was combined with the present SPAMS, it could achieve efficient focusing transmission of 120 nm-10 μm PM, and the experimental results were remarkably consistent with the theoretical predictions.

[Figure]

Fig. 1 Schematic diagram of the total wide range aerosol sampling system

[Figure]

Fig. 2 Detection results of particle transmission efficiencies by SPAMS

- At these high vacuum aerodynamic diameters, what is the laser hit rate? (i.e. what proportion of optically-detected particles produce a mass spectrum?)

**Answer:** This is really a very professional question, thank you very much. The laser strike rate of different bacteria is different, as shown in the following table. The laser hit rate is related to the shape of the bacteria itself and the injection concentration.

Table 1 Laser hit rate of different strains

| Name | Hit rate |
| --- | --- |
| Klebsiella pneumoniae | 25.17% |
| Salmonella pneumoniae | 33.77% |
| Shiga virulent Escherichia coli | 45.77% |
| Bordetella bronchitis | 21.62% |
| Escherichia coli | 24.71% |
| Staphylococcus aureus | 62.85% |
| Listeria monocytogenes | 32.38% |
| Enterococcus faecium | 54.11% |
| Enterobacter cloacae | 40.83% |
| Staphylococcus epidermidis | 17.73% |
| Candida albicans | 54.26% |
| Candida tropicalis | 13.25% |
| Candida glabrata | 32.37% |
| Aspergillus brasiliensis | 15.87% |
| Saccharomyces cerevisiae | 40.57% |

- p. 7, lines 178-193: It is not entirely clear if this paragraph describes previous investigators' results or your own. I found the sentence in lines 182-185 especially difficult to understand.

**Answer:** Lines 182-185 show the results of our study, which are compared with those of the same type of mass spectrometer in the world. We found the description unclear and have since revised it. Thank you very much for your revision.

**"Czerwieniec et al. (2005) found similar peaks when detecting vegetative cells of Bacillus atrophaeus and speculated that +30, +70, +72, +74, +86, +110, and +120 were decarboxylic ionic peaks of amino acids. Srivastava et al. (2005) speculated that +59, +81, +84, and +88 ionic peaks were organic fragments containing nitrogen. Zeng et al. (2019) used SPAMS to detect 13 strains of bacteria to obtain similar bioaerosol characteristic ions; however, there were fewer characteristic peaks in the negative mass spectrum, and no negative ions with a mass charge ratio greater than 200 were detected, and the overall ionic peak signal was weak. HP-SPAMS detected the mass spectra of 15 strains of pure bacteria as shown in Fig. 3. Fig. 3 shows the stacked mass spectrograms of 14119 biological single particles with mass number of -300-300 Da ion peaks. In addition to $^{23}Na^+$ and $^{39}K^+$ metal ion peaks, there are also a large number of amino acid decarboxylic ion peaks in the positive spectrum. The positive ion peaks were mainly $^{30}[Glycine-COOH]^+$, $^{59}[C_3NH_9]^+$, $^{70}[Proline-COOH]^+$, $^{72}[Valine-COOH]^+$, $^{74}[Threonine-COOH]^+$, $^{84}[C_5NH_{10}]^+$, $^{86}[Leucine-COOH]^+$, $^{110}[Histidine-COOH]^+$, and $^{120}[Phenylalanine-COOH]^+$. The negative ionic peaks were mainly organic nitrogen $^{26}CN^-$, $^{42}CNO^-$, phosphate $^{63}PO_2^-$, $^{79}PO_3^-$, $^{97}H_2PO_4^-$, $^{159}H(PO_3)_2^-$, $^{199}NaH_2P_2O_7^-$, and other common**

**biological ionic peaks."**

- Figure 3: I'm not sure I understand what is indicated by the different colors in this figure.

**Answer:** Fig.3 shows the mass spectrograms of 14119 biological single particles stacked with ion peaks whose mass number is -300~300 Da. Each color represents the proportion of the signal intensity range of an ion peak area, which can show the mass spectral peak with strong signal more intuitively. For clearer expression, we have added the annotated information.

[Figure]

Fig. 3 Stacking diagram of the area of all bioaerosol ion peaks; (a) Positive mass spectrum; (b) Negative mass spectrum

- It would be useful to include some discussion of method uncertainties in the paper. What are the limiting factors in translating the results of this lab-based study to the atmosphere?

**Answer:** Thanks for your suggestions. We added this part. The classification method based on the ratio of ion peaks based on HP-SPAMS can effectively distinguish bioaerosols from abiotic aerosols. However, due to the difference in the signal intensity of the four characteristic ion peaks, not all particles contain four ion peaks at the same time, and it is impossible to distinguish 100% biological particles. In addition, the classification method using only the data measured in the laboratory cannot be directly applied to the classification of environmental particles. Bioaerosols undergo aging reactions in the atmospheric environment, and their chemical composition and morphology characteristics have different changes under the influence of a variety of natural factors. Moreover, it is necessary to consider all the mass spectrometric characteristics to identify environmental particles.

---

## Author Comment (AC3)

**Reply on RC3**

Review of "Technical Note: Bioaerosol identification by wide particle size range single particle mass spectrometry" by Li et al.

The authors present a method to differentiate laboratory-generated bioaerosol particle samples from other particles that exhibit similar ion signatures in a single-particle mass spectrometer. The authors cite recently published improvements to their instrument, including sampling size range and ion extraction. The authors include a useful analysis on how ionization laser power affects critical signals for bioaerosol identification.

Generally, the paper is not very well written and is difficult to follow. It is unclear how the experiments were actually performed, how the analysis generated the conclusions, and how the presented method would perform under realistic atmospheric conditions or against similar published methods.

The paper is not publishable in its present form. Significant improvements must be made in a variety of areas, specified below as Major and Minor Comments. The underlying method and results appear to have scientific value, but the authors must first present them clearly and completely.

Major Comments

Results and methods lack critical detail and context with previous studies.

- 1a) The references and descriptions of other single-particle mass spectrometers and previous work on bioaerosol identification are inappropriate, out of date, or too limited in scope. Add a paragraph to the Introduction describing some of the previous bioaerosol identification studies, particularly those involving online mass spectrometers, and perhaps also mentioning other successful techniques (fluorescence, Raman, offline methods). These references are suggested starting points only, and the authors should choose appropriately.

Huffman et al., Real-time sensing of bioaerosols: Review and current perspectives, Aero Sci Tech. 2020, doi: 10.1080/02786826.2019.1664724

Russell et al., Microorganism characterization by single particle mass spectrometry, Mass Spec Rev, 2008 https://doi.org/10.1002/mas.20198

Pratt and Prather, Mass spectrometry of atmospheric aerosols—Recent developments and applications. Part II: On-line mass spectrometry techniques, Mass Spec Rev, 2011, https://doi.org/10.1002/mas.20330

Huffman and Santarpia, Online Techniques for Quantification and Characterization of Biological Aerosols, Microbiology of Aerosols chapter 1.4, 2017, https://doi.org/10.1002/9781119132318.ch1d

**Answer:** Thank you very much for your suggestions. We discussed fluorescence detection of bioaerosols in the introduction, and we have now appropriately added two other successful detection techniques (Raman and offline methods).

- Also, the paper references experimental aerosol studies and inlets without identifying the specific instruments in the text. Specify to which instrument (ATOFMS, AMS, PALMS, etc) the publications refer, e.g., in lines 61, 69-78, and elsewhere. Lastly, the method presented by the authors identifies bioaerosol particle samples using ion ratios of $PO_3^-/PO_2^-$ and $CNO^-/CN^-$, refined with machine learning. This method exactly follows that of Zawadowicz et al., 2017 using the PALMS single-particle instrument. Although the authors do include this reference in a brief sentence (line 61), they should state (e.g., in the final para of section 1) that the analysis method of the current study is based on Zawadowicz. Also consider many relevant ATOFMS publications and their use of these ions or ion ratios.

**Answer:** Thank you very much for your suggestions. We have added the instruments of the reference publication in the corresponding position. In addition, the final paragraph of section 1 is modified to clearly show that the analysis method of this research is based on the development of Zawadowicz.

- 1b) The authors' instrument is inadequately described (section 2.1). In addition to the Li et al. 2011 reference, describe the instrument details such as detection and ionization lasers, previous size range and detection efficiency, and any other characteristics relevant to the current work. Since section 3 discusses spectral variation due to ionization energy, a typical laser beam width would be helpful. How does this instrument compare to previous single-particle mass spectrometers used in bioaerosol detection (ATOFMS, PALMS, SPLAT, others). What type of time-of-flight mass spectrometer does SPAMS employ (a commercial model?). How similar is SPAMS to the commercial ATOFMS?. State clearly what differentiates SPAMS from the "high-performance" version used in this study. Define "pore size". What is "multi-channel superimposed signal acquisition system"?

**Answer:** Thank you very much for your suggestions. We supplement the details of the

HP-SPAMS used in Section 2.1, including the sizes and models of the sections.

- 1c) The performance of the new instrument SPAMS configuration is presented without context to similar instruments' performance on bioaerosol detection. Specifically, how do the discrimination percentages presented here compare to those in the literature? Choose similar aerosol systems if possible, and/or list limitations in the comparisons. Direct comparisons to Zawadowicz seem obvious.

**Answer:** Thank you for your suggestions. SPAMS have a similar aerosol system. We have made a direct comparison between the structure of the new instrument HP-SPAMS and PSLMS, and the performance indexes are briefly described in the following table.

Cziczo et al., Particle analysis by laser mass spectrometry (PALMS) studies of ice nuclei and other low number density particles, International Journal of Mass Spectrometry 258 (2006) 21–29, https:// doi:10.1016/j.ijms.2006.05.013

Thomson, D. S., Schein, M. E., and Murphy, D. M.: Particle analysis by laser mass spectrometry { WB}-57 instrument overview, Aerosol Sci. Technol., 33, 153–169, 2000.

Murphy, D. M., and Thomson, D. S. Laser Ionization Mass Spectroscopy of Single Aerosol Particles, Aerosol Sci. Technol. 22:237-249,1995.

Table 1 A brief comparison of structural performance between HP-SPAMS and PSLMS

|  | PALMS | HP-SPAMS |
| --- | --- | --- |
| Sample structure | isobaric (~40 mb) aerodynamic inlet | 7-stage aerodynamic lens |
| Laser caliper | 532nm Nd:YAG | 405nm Nd:YAG |
| Ionization laser | 193nm excimer laser | 266 nm, Nd: YAG laser |
| Mass analyzer | unipolar reflectron time-of-flight mass spectrometer | Bipolar time-of-flight mass spectrometer (Z-TOF) |
| Particle size transmission range | ~150 nm to 2.0 μm | ~150 nm to 10.0 μm |
| Mass resolution (the half-peak width method, m/△m) | ~300 | ~2000 |

Conclusions are not supported by the data as presented.

- 2a) A principal conclusion of the study is that "The ionized laser energy has a certain influence on the integrity of the ionic peak but hardly affects the identification accuracy of bioaerosols." (line 320). Specifically, line 284 states, "…the discrimination degree of bacterial aerosols and dust under 0.5, 0.75, 1.0, 1.25, and 1.5 mJ energies were 96.6%, 97.4%, 97.1%, 96.5%, and 97.8%, respectively, indicating that the ionized laser had little effects on

discriminating biological aerosols and dust disruptors." However, in apparent contradiction to this constant discrimination efficiency, which is based on $PO_3^-/PO_2^-$ and $CNO^-/CN^-$ peak ratios, Figure 8 seems to indicate that most dust spectra (~70% or so) do not contain either phosphate peak. I interpret Figure 8 as plotting occurrence frequency of these peaks, not "peak ratio%" as listed in the y-axis. Clearly describe how the method can differentiate between dust and bioaerosol when a large fraction of dust spectra are apparently excluded from the analysis due to missing peaks. State what fraction of each particle type sample is excluded from the analysis prior to applying the classification routine. Given this apparent limitation, how would the authors' technique be used realistically on an externally mixed population of particles with unknown composition?

**Answer:** Thank you very much for your suggestions. The hard ionization mode of single particle mass spectrometry can produce different degrees of ion debris during the process of particle desorption, especially at high laser energy. The resolutions of bioaerosol and dust interferors at five different laser energies were 96.6%, 97.4%, 97.1%, 96.5% and 97.8%, indicating that the resolutionwas definitely above 95% as long as the four characteristic peaks were present and the classification method was used. Fig. 8 shows the frequency of ion peaks. Fig. 4 shows that dust particles were excluded from the analysis because 50.5% of the particles lacked characteristic peaks. We used the characteristic peak ratio method to exclude dust particles in the process of identifying bioaerosol, and most dust particles were excluded from the analysis because of the lack of four characteristic peaks to fundamentally avoid causing interference. When attempting to discriminate dust in atmospheric datasets, the entire mass spectrum characteristics must be considered. We should still use established signatures and ion markers to identify dust particles. On this basis, a supplementary method for identifying dust and biological particles is proposed.

- 2b) The authors report using a supervised machine learning algorithm to help differentiate bioaerosol and abiotic aerosol, claiming a 97.7% accuracy. This successful discrimination is the principal conclusion of this study. However, the authors only mention the technique in passing, as a single sentence in section 3.3. Provide details of the machine learning algorithm and relevant parameters in a separate paragraph. Give enough information that another group could recreate these results. How is the training dataset defined? What is the test dataset? How many spectra were used in the analysis? How many were rejected?

**Answer:** Thank you very much for your suggestions. We supplement the details of the machine classification algorithm used in Section 3.3. The data analysis in this case is based on the Computational Continuation Core (COCO, V1.3), cubic SVM algorithm is implemented based on Statistics and Machine Learning Toolbox (Statistics and Machine Learning Toolbox 11.2) in MATLAB 2017b (Classification Learner), where PCA was 95%. Train models to classify data

using supervised machine learning. A random 30% dataset is used as the training set, and the empirically determined nonlinear kernel functions can provide the best performance in this case. All particles with four characteristic ion peaks were analyzed, as shown in Table 3, and 82.9 bacterial aerosols and 52.8% fungal aerosols could be distinguished.

Inadequate presentation of material.

   3a) A large fraction of the paper is written in a way that is vague, redundant, or unclear. Critical information is missing or lost. The sentence structure, writing clarity, and grammatical accuracy need significant improvement prior to publication.

Examples include…

- Line 87 from the Intro:

"The analysis of single particle mass spectra is a hard ionization process and laser energy has little effect on the discrimination of this classification method."

**Answer:** We changed the last paragraph of the first paragraph to avoid some vague statements.

- Line 309 from the Conclusion:

"The performance of SPAMS and the improvement of the sampling system have improved the ability to identify bioaerosols."

**Answer:** "The performance of SPAMS and the improvement of the sampling system have improved the ability to detect bioaerosols."

- Lines 317-320 from the Conclusion:

"In addition, due to the influence of laser ionization efficiency, the effective mass spectra peak ratio of bacterial aerosol generation is higher, thus it is more suitable for this method. The ionized laser energy has a certain influence on the integrity of the ionic peak but hardly affects the identification accuracy of bioaerosols."

**Answer:** "In addition, due to the influence of laser ionization efficiency, the effective mass spectra peak produced by bacterial aerosol is higher than 80%, which is more suitable for this method. The changes of $PO_3^-/PO_2^-$ and $CNO^-/CN^-$ values at different laser energies show that the ionized laser energy affects the integrity of particles, but does not affect the identification accuracy based on the characteristic peaks of bioaerosols."

- There many examples throughout the paper. The authors edit the paper again for proper sentence structure, clarity, verb conjugation, plural nouns, and definite and indefinite articles. If necessary, employ an English language editing service. Consider an alternative term for "disruptors" to describe abiotic particles.

**Answer:** Thank you very much for your advice. We went through the manuscript and revised it.

- 3b) The acronym "SPAMS" is used confusingly to describe both single-particle mass spectrometers in general (eg, ATOFMS, PALMS, etc), and also the specific instrument used by the authors in their experiments. Choose unique acronyms to describe other single-particle mass spectrometers. Note also that the Aerodyne AMS is not a single-particle mass spectrometer.

**Answer:** Thanks. The Aerodyne AMS, which we refer to in line 72, mainly refers to the aerodynamic lens technology of this instrument. For non-single-particle mass spectrometers, we do not use the acronym "SPAMS" in the manuscript.

Minor Comments

- Line 41: The sentence seems out of place. What makes identification unclear? Their scattered sources? Also, the Rosch 2006 reference study is not appropriate for this statement. There are dozens of papers describing bioaerosol detection subsequent to this study.

**Answer:** Line 41 describes the difficulty in identifying the sources of the bioaerosols. We have removed inappropriate points and described other methods for detecting biological particles in the Introduction section.

- Line 44: Consider adding other reviews of bioaerosol identification, eg, Huffman et al., Aero Sci Tech, 2020.

**Answer:** Thank you for your good suggestion. We have added relevant reviews.

- Line 48: add references for mineral fluorescence

**Answer:** Pulsed laser excitation of the mineral samples at 355 and 266nm often resulted in strong fluorescence.

Bozlee, B. J., Misra, A. K.,Remote Raman and fluorescence studies of mineral samples, Spectrochimica Acta Part A Molecular & Biomolecular Spectroscopy, 61(10): 2342-2348, 2005.doi: 10.1016/j.saa.2005.02.033

- Line 65.   I suggest you describe why 98% discrimination (line 64) is "insufficient".

**Answer:** We are sure that "98%" is a high degree of discrimination. In the process of literature review, we found that there are few studies on the distinction between fungi and abiotic aerosols. Here, the sentence was changed as "**However, there is insufficient research on the detection and differentiation of other bioaerosols such as fungi from abiotic aerosols**".

- Line 69.   Add a reference for the typical particle size range.

**Answer:** Leone, N., Descroix, D., Mohammed, S., Bioaerosol Detection Technologies, 143-167, 2014. ISBN: 978-1-4419-5581-4.

- Line 78 seems out of context. Please remove or clarify. Define ATOFMS.

**Answer:** We rephrased it.

- Line 113: Why do you classify these aerosol as "disruptors"?

**Answer:** Based on the widely used fluorescence technology, the interference existed in the process of bioaerosol was identified, and the frequency of the characteristic mass spectral peak of the organism was higher than 50%. In this paper, automobile exhaust, biomass combustion products and road dust are defined as " disruptors ".

- Line 114: What kind of "road dust" did you use in this study? Is it a commercial sample?

**Answer:** "road dust" is "Guangzhou Accelerator Industrial Park road dust". We have added to the manuscript a discussion of specific types of abiotic nitrogen and phosphorus distractives.

- Line 127: "absorbed" seems incorrect here

**Answer:** "absorbed" is replaced by "removed".

- Line 138: "A sheath gas of 80 kPa of clean air was used." I don't understand the pressure of "sheath" gas here. Reword for clarity, eg, "a ## flow of dilution air…".

**Answer:** This sentence is not clearly stated, 80 kPa refers to the aerosol generator pressure indicator number.

- Table 1.   List the type of bioaerosol, bacteria or fungi.

**Answer:** We have added the types of bioaerosols in Table 1.

- Figure 1.   Suggest experimental "design" or "configuration" rather than "flow".   Where is the "exhaust port with a high-efficiency particulate air filter"?

**Answer:** Thank you for your good suggestion. We added "exhaust port with a high-efficiency particulate air filter" to Fig. 1.

- Fig 2. Which samples are bacteria?   Which are fungi?

**Answer:** The numbers in Fig. 2 correspond to those in Table 1. #1 to #10 are bacteria samples and #11 to #15 are fungi samples.

- Section 3.1. The authors compare the size of bioaerosol as detected by SPAMS, which like all single-particle mass spectrometers has size-dependent counting biases, with electron microscopy size distributions. Although the relative comparisons of aerosol sizes in this section remain valid, the authors should make it clear that the "overall particle size distribution" is the size as detected by SPAMS and not an absolute size distributions of the aerosol samples. The related statements of lines 170-173 need clarification. Do these statements refer to SPAMS, or to single-particle mass specs in general…?

**Answer:** The "overall particle size distribution" is the size as detected by SPAMS, not an absolute size distributions of the aerosol samples. We have added a description on lines 170-173.

- Line 181 & Fig 3. Units?

**Answer:** Units are mV. In order to express this conclusion more clearly, we have modified the figure and text of this paragraph.

- Line 185. With what instrument?

**Answer:** Bioaerosol mass spectrometry (BAMS).

- Line 198. Clarify "speculated and added".

**Answer:** Russell et al. and Czerwieniec et al. believed that m/z -277 was $Na_2H(PO_3)_2(PO_4)^-$, -261 and -277 were first obtained by HP-SPAMS detection of bioaerosols. We speculated and added that -277 was $NaH(PO_3)_2(PO_4)^-$.

Russell et al., Microorganism characterization by single particle mass spectrometry, Mass Spec Rev, 2008 https://doi.org/10.1002/mas.20198

Czerwieniec,G. A., Russell, S. C., Tobias, H. J., Pitesky, M. E., Fergenson, D. P., Steele, P., Srivastava, A., Horn, J. M., Frank, M., Gard, E. E., and Lebrilla, C. B., :Stable Isotope Labeling of Entire Bacillus atrophaeus Spores and Vegetative Cells Using Bioaerosol Mass Spectrometry, Anal. Chem., 1081-1087, https://doi.org/10.1021/ac0488098, 2005.

- Line 199. SPMS is undefined.

**Answer:** SPMS is Single particle mass spectrometry.

- Line 215-218. The selection criteria are unclear. Does "alone" mean one of those 4 individual peaks? Does "interference" mean the spectrum contains one of those peaks?

**Answer:** "Alone" is not an accurate word. In the manuscript, we define "disruptors". "Interference" mean the spectrum contains those 4 individual peaks.

- Line 225-226. Redundant

**Answer:** Thank you for your good suggestion. We have removed it.

- Fig 4. The y-axis label seems incorrect

**Answer:** The y-axis label is occurrence frequency of peaks. We have modified Fig. 4.

- Line 230-231. These numbers don't correspond to anything in particular in Fig 3. "proportion interval"…? "concentrated"…?

**Answer:** Lines 230 and 231 describe the scatter distribution of bioaerosol and abiotic aerosol in Fig. 5.

- Line 236. Add references for the "traditional method"

**Answer:** The traditional method in the manuscript refers to the characteristic ion markers.

- Line 248. How "high"?

**Answer:** For lines 248 and 249, we rephrase as "The discrimination method based on the characteristic peak ratio had higher identification rate for bacterial aerosols than fungal aerosols".

- Line 249-251. These morphology sentences are out of context in this para. Remove or add text to describe their relevance.

**Answer:** The morphology sentences of bacteria and fungi are to show that the shape and structure of the particles are related to the laser ionization efficiency. We explain it on line 250.

- Line 257. "morphology of organic compounds" ?

**Answer:** "composition of organic compounds".

- Line 267. "peak integrity"?

**Answer:** "particles integrity"

- Line 290-304 and Fig 8. Clarify and use consistent terminology. Should "peak output rate" and "peak ratio" actually refer to occurrence frequency of peaks? The % values do not obviously correspond to any consistent set of points in Fig 8. Clarify/correct these values.

**Answer:** "peak output rate" and "peak ratio" actually refer to occurrence frequency of peaks. We corrected the % value to correspond to any consistent set of points in Fig 8.

- Data Availability. Include a publicly accessible link to data prior to publication.

**Answer:** Thank you for your good suggestion. We created a accessible link to data.

https://pan.baidu.com/s/1KHGEGYQZA0_XuPOozpYdBw          code: 9A7U